# AutoML in the Age of Large Language Models: Current Challenges, Future Opportunities and Risks

**Alexander Tornede**                                        *a.tornede@ai.uni-hannover.de*

**Difan Deng**                                               *d.deng@ai.uni-hannover.de*

**Theresa Eimer**                                           *t.eimer@ai.uni-hannover.de*
*Institute of Artificial Intelligence, Leibniz University Hannover*

**Joseph Giovanelli**                                        *j.giovanelli@unibo.it*
*Alma Mater Studiorum – University of Bologna*

**Aditya Mohan**                                            *a.mohan@ai.uni-hannover.de*

**Tim Ruhkopf**                                             *t.ruhkopf@ai.uni-hannover.de*

**Sarah Segel**                                              *s.segel@ai.uni-hannover.de*
*Institute of Artificial Intelligence, Leibniz University Hannover*

**Daphne Theodorakopoulos**                          *d.theodorakopoulos@ai.uni-hannover.de*
*Institute of Artificial Intelligence, Leibniz University Hannover, German Research Center for Artificial Intelligence (DFKI)*

**Tanja Tornede**                                            *t.tornede@ai.uni-hannover.de*
*Institute of Artificial Intelligence, Leibniz University Hannover*

**Henning Wachsmuth**                                       *h.wachsmuth@ai.uni-hannover.de*

**Marius Lindauer**                                         *m.lindauer@ai.uni-hannover.de*
*Institute of Artificial Intelligence, L3S Research Center, Leibniz University Hannover*

**Reviewed on OpenReview:** *https://openreview.net/forum?id=cAthubStyG*

## Abstract

The fields of both Natural Language Processing (NLP) and Automated Machine Learning (AutoML) have achieved remarkable results over the past years. In NLP, especially Large Language Models (LLMs) have experienced a rapid series of breakthroughs very recently. We envision that the two fields can radically push the boundaries of each other through tight integration. To showcase this vision, we explore the potential of a symbiotic relationship between AutoML and LLMs, shedding light on how they can benefit each other. In particular, we investigate both the opportunities to enhance AutoML approaches with LLMs from different perspectives and the challenges of leveraging AutoML to further improve LLMs. To this end, we survey existing work, and we critically assess risks. We strongly believe that the integration of the two fields has the potential to disrupt both fields, NLP and AutoML. By highlighting conceivable synergies, but also risks, we aim to foster further exploration at the intersection of AutoML and LLMs.

## 1 Introduction

Large Language Models (LLMs) (Zhao et al., 2023a) are currently on everybody's lips due to the recent series of rapid breakthroughs achieved, such as self-attention (Vaswani et al., 2017), BERT (Devlin et al., 2019), several versions of GPT (Radford et al., 2018; 2019; Brown et al., 2020; OpenAI, 2022; 2023), LaMDA

(Thoppilan et al., 2022), LLaMA (Touvron et al., 2023), or OpenAssistant (Köpf et al., 2023). The term LLM refers to a language model where the actual model is instantiated by a deep neural network that typically features millions to billions of weights.[1] Such LLMs are pre-trained on extremely large corpora of textual datasets. Due to their excellent capabilities on various Natural Language Processing (NLP) tasks, they have the potential to lead to the democratization of NLP, if their pre-trained versions are accessible to the broad public and the power of LLMs does not lay in the hand of a few companies with sufficient financial resources. Additionally, we highlight the emerging possibilities of multimodal LLMs (Yin et al., 2023; Wu et al., 2023b). By incorporating various data modalities, such as audio or images, these models enable a more flexible communication with users by capturing information presented in non-textual formats as well as facilitating the output of information in non-text formats.

Similarly, Automated Machine Learning (AutoML) (Hutter et al., 2019) democratizes Machine Learning (ML) by supporting data scientists in finding well-performing ML pipelines for specific tasks through (partial) automation. AutoML has achieved remarkable success over the last decade with heavily-used open-source frameworks such as Auto-WEKA (Thornton et al., 2013; Kotthoff et al., 2017; 2019), AutoSklearn (Feurer et al., 2015a; 2019; 2022), AutoGluon (Erickson et al., 2020), Auto-PyTorch (Zimmer et al., 2021), and closed commercialized frameworks.

With this paper, we want to highlight our vision in which AutoML and LLMs are integrated with each other to radically push the boundaries of both AutoML and NLP. On the one hand, we expect that applying AutoML to LLMs improves various stages of the LLM lifecycle by increasing their capabilities and making them more efficient. On the other hand, the disruptive NLP and knowledge-modeling capabilities of LLMs can unleash the full potential of AutoML both via an integration as a human-machine-interaction component, and as a technical meta-learning component within AutoML frameworks themselves. Correspondingly, this survey is targeted both 1) at NLP researchers leveraging AutoML methods to further improve LLMs and 2) at AutoML researchers who seek to leverage the strengths of LLMs to improve AutoML paradigms and tools in various regards outlined below. Consequently, we only sparsely mention general topics or problems concerning LLMs, but focus on problems and aspects that arise from the intersection of the two fields. For a more general survey on LLMs, we refer the reader to Zhao et al. (2023a). To make this paper as tailored as possible, we explicitly chose not to focus on pre-trained models in general, but focus on language models as they allow extracting knowledge from the large set of unstructured data in the form of text on the web which certainly contains valuable AutoML knowledge.[2]

To showcase our vision, we explore the potential of a symbiotic relationship between AutoML and LLMs, including a survey of existing work on the matter. We start by investigating the challenges of applying AutoML to LLMs, in particular Neural Architecture Search (NAS) (Elsken et al., 2019; Wistuba et al., 2019; White et al., 2023) and Hyperparameter Optimization (HPO) (Feurer & Hutter, 2019; Bischl et al., 2023), as well as potential solutions for optimizing pre-training, fine-tuning, and inference (Sec. 2). Subsequently, we swap perspectives and elaborate on opportunities offered by LLMs to improve AutoML approaches in terms of human-machine-interaction, the configuration of AutoML systems, and the replacement of parts of AutoML systems by an LLM (Sec. 3). In order to give a balanced outlook, we also critically assess potential risks which might arise due to an integration of AutoML and LLMs (Sec. 4).

The main insights we want to bring across with this work are the following:

(i) Current AutoML approaches are not ready for a holistic optimization of the whole lifecycle of LLMs due to several challenges, such as the computational demand of pre-training and the multi-stage training process of LLMs featuring varying metrics and learning paradigms.

(ii) An integration of LLMs with AutoML tools has the potential to substantially improve the human-machine-interaction component of corresponding tools, alleviate the tedious task of correctly configuring an AutoML tool and to improve several internal components of AutoML tools through knowledge gained on meta-learning from unstructured data.

---

[1]We note that there is no clear threshold after which a language model is called large.

[2]We note that there also exists work in the broader intersection of AutoML and pre-trained models (Hollmann et al., 2023a; Müller et al., 2023) which is outside the scope of this paper as we only focus on (pre-trained) LLMs and not pre-trained models in general.

(iii) Integrating the two research areas with each other naturally also bears risks such as inadequate evaluation of LLM powered AutoML systems, catastrophically wrong behavior of AutoML systems due to hallucinations of an LLM powered component, (too) high trust in results of an AutoML tool explained through natural language and ever-increasing resource demands.

## 2 AutoML for LLMs

One could argue that AutoML for LLMs is yet another application of AutoML. However, compared to previous applications of AutoML to different learning paradigms, LLMs come with a complete set of new challenges, rendering the standard approaches of existing AutoML tools partially useless. In fact, Godbole et al. (2023) argues that standard HPO tools cannot be applied off-the-shelf for optimizing very deep and thus resource-intensive neural networks such as LLMs. Furthermore, as pointed out by Hutter (2022) in his vision for Deep Learning 2.0, we need to bring many ideas together and face new challenges if we want to automatically obtain models of the same or a better quality level as the ones currently manually designed for deep learning including LLMs.

In a nutshell, we see five main challenges which need to be addressed:

  (i) Pre-training the base model of LLMs is extremely expensive such that very few full training runs – maybe even only a single one – of the LLM base model are possible.

 (ii) The AutoML task is complex and ideally has to be performed across many steps of the LLM lifecycle, including pre-training, several steps for fine-tuning, and inference. However, these steps currently cannot be addressed in a single joint, holistic optimization and instead have to be performed independently most of the time;

(iii) Finding good neural architectures to solve a specific problem is a tedious task and modern automated methods such as NAS can only do so to a certain extent, but have not yet been able to produce new ground-breaking architectures.

(iv) All of the stages of the LLM lifecycle call for the optimization of a different metric. Especially in pre-training this metric can be seen to act as a proxy for the final performance across a variety of different tasks, which the LLM could be used for. This can lead to potentially misleading, noisy, and biased performance indicators for the AutoML process.

 (v) AutoML commonly considers only a single learning paradigm (e.g. supervised learning) at a time. However, training LLMs is challenging as the different stages of the lifecycle require different learning paradigms.

After providing more background on LLMs in the next subsection, we delve into details for all these challenges and discuss existing AutoML ideas that are either applied already to LLMs or could be a promising future direction.

### 2.1 Background on LLMs

The lifecycle of an LLM typically involves three key stages: pre-training, fine-tuning, and inference (Zhao et al., 2023a). Each stage of this lifecycle comes with its own set of objectives, hyperparameters, and design decisions affecting the final performance on downstream tasks, as depicted in Figure 1.

During **pre-training**, the base model is trained on a large corpus of unlabeled text to learn language patterns and capture general knowledge about the language. The overall goal of pre-training is to produce useful representations of sequences with a pre-text task in a quasi-unsupervised fashion which corresponds to the first stage of the Self-Supervised Learning (SSL) paradigm (Balestriero et al., 2023). It is typically the most expensive part of training an LLM and only a few groups and companies world-wide can afford to pre-train a base model.

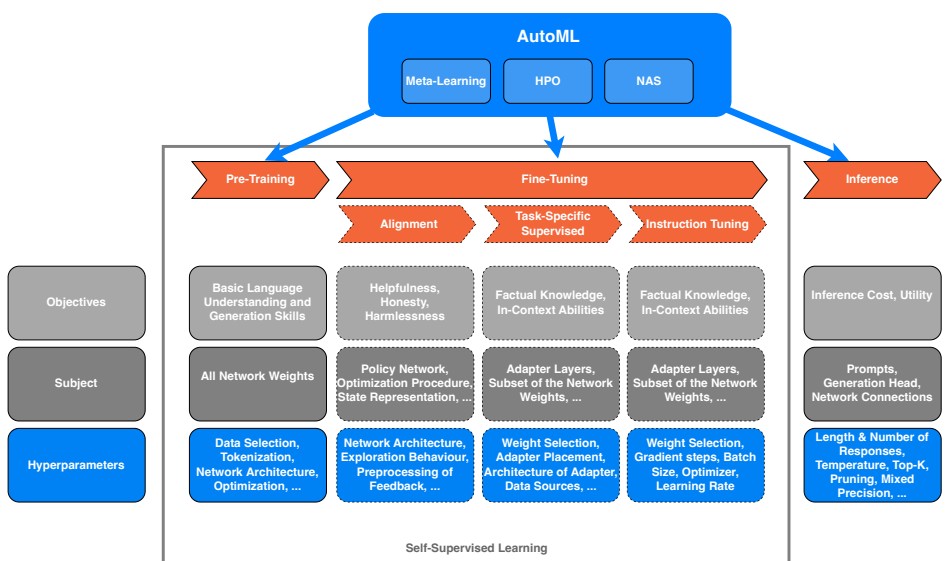

Figure 1: AutoML can be used in all stages of the LLM lifecycle and needs to be adjusted to the different objectives, hyperparameters, and design decisions of each stage. The graphic depicts exemplary objectives, subjects of optimization, and associated hyperparameters. Due to computational constraints, the stages are considered separately, one after the other.

**Fine-tuning** follows, where the pre-trained model is further trained on domain- or task-specific data to specialize its knowledge and adapt to specific tasks, domains, or use cases. In general, fine-tuning covers various groups of approaches such as task-specific supervised fine-tuning (Zhao et al., 2023a), instruction tuning (Wei et al., 2022), and alignment fine-tuning (Zhao et al., 2023a). The success in this specialization is quantified with a corresponding metric or loss function. Provided the pre-training found useful representations, the objective here is to utilize or adapt these representations with specific downstream tasks in mind. This can be very effective even with limited data for the downstream task and thus is computationally much less demanding than training a base model. The probable reason is that for a given downstream task, there often exists a low-dimensional reparametrization of the strong joint embedding which was learned during pre-training featuring a large number of parameters. This allows to unleash the full power of transfer learning (Aghajanyan et al., 2021).

Instruction tuning can be seen as generalized task-specific supervised fine-tuning that teaches LLMs to follow instructions in the form of prompts. It fine-tunes a single model using instructions via prompts that span various tasks, thereby enabling the prompting of LLMs. A very specific and, from an AutoML perspective, complicated task that is often solved by a form of instruction tuning, called Reinforcement Learning from Human Feedback (RLHF) (Fernandes et al., 2023), is alignment.[3] Alignment (fine-tuning) describes the idea of approximately aligning the behavior and output of LLMs with human value-related task objectives such as helpfulness, honesty, harmlessness, etc. The two ingredients required to make Reinforcement Learning (RL) work on a pre-trained Language Model (LM) are: (i) A Reward Model (RM) to convert a sequence of texts to a scalar reward which numerically represents the human preference. (ii) A policy that takes user queries or prompts as input to produce language tokens as outputs.

We note that, in principle, one can also use a pre-trained model off-the-shelf. However, in practice, some instruction tuning and alignment fine-tuning is performed if the model is released to end-users. Moreover, some form of additional task-specific fine-tuning is often performed to increase the performance of the LLM for its designated usage and to make it properly usable.

---

[3]We note that there are recent approaches aimed at replacing reinforcement learning with supervised approaches such as classification losses (Rafailov et al., 2023). Moreover, in general instruction tuning covers a large range of techniques such as self-instruct (Wang et al., 2023b).

Finally, during **inference**, the fine-tuned LLM generates text for language-related tasks, such as question answering, based on the learned knowledge and contextual understanding.

## 2.2 Challenge I: Cost of Pre-Training Base Models

Applying standard black-box AutoML approaches, such as grid search, random search, Bayesian Optimization or evolutionary algorithms, to pre-train LLMs is simply not feasible since a single pre-training of an LLM requires hundreds of GPUs for days. Brown et al. (2020) estimate that a single training of GPT-3 required months on a thousand V100 GPUs. To put it in numbers, consider tuning 10 hyperparameters and that a black-box AutoML optimizer requires at least 10 times the number of hyperparameters as samples [4], we would need 100 months (i.e., more than 8 years) on a thousand V100 GPUs. Even when considering smaller and potentially open-source LLMs, many still require training on a considerable amount of compute units (be it GPUs or TPUs) for several days, yielding training resource requirements of hundreds of exaFLOPS (Geiping & Goldstein, 2023) just for a single training run. Since emerging abilities of LLMs only happen from a certain size onward, size is important and thus, large training costs are often unavoidable. In view of recent state-of-the-art approaches to AutoML, there could be two ways to address this, as discussed in the following.

### 2.2.1 Prior-Guided Multi-Fidelity Optimization with Scaling Laws

In cases where many full training evaluations with different hyperparameters or neural architectures are not feasible, a common AutoML approach is multi-fidelity optimization. The general idea is to approximate the real target function by a less expensive fidelity (e.g., by training for fewer epochs or by training on less data) making this a natural candidate strategy for HPO for LLMs.

However, even multi-fidelity approaches require at least tenths of full training runs to perform well. At the same time, human experts are somewhat successful in tuning LLMs manually without excessive re-trainings. Guided by this observation on earlier AutoML problems, the community developed approaches leveraging intuitive prior knowledge of experts. While Souza et al. (2021) and Hvarfner et al. (2022) proposed ways to integrate prior knowledge on the location of promising hyperparameter configurations into Bayesian Optimization, Mallik et al. (2023) extended this idea to multi-fidelity optimization and achieved strong performance within ten full training runs.

Moreover, multi-fidelity approaches have to be configured correctly, in particular regarding the choice of the fidelity type, to achieve strong performance. This choice should be based on the observation that the ordering among potential configurations on low-fidelity approximations should be correlated with the maximum fidelity. It is currently unclear, if such an assumption can be made for LLMs considering recent work, e.g., by Kirsch et al. (2022). Even more related, in the pursuit of understanding the interplay between the size of a language model and its performance, recent works have delved into the scaling laws of language models (Radford et al., 2019; Brown et al., 2020; Kaplan et al., 2020). They showed that improvements along scale dimensions generally lead to an increase in performance, both for parameter scaling dimensions such as the network width and depth, computational scaling dimensions such as the number of training steps, and the amount of data used for training. When not limited by the other two factors, performance exhibits a power-law correlation with each of the three scale factors, bounded by diminishing returns. Correspondingly, under reasonable assumptions, multi-fidelity approaches seem to be indeed very suitable, if correctly configured.

Motivated by the same idea of leveraging cheap approximations, Yang et al. (2021) mitigate the large cost of LLM hyperparameter optimization by leveraging specific network parametrizations that allow for stable training across model sizes. This way, a smaller version of the actual model can be tuned, and the best-found hyperparameter configuration can be transferred to the larger model. However, the approach is limited to hyperparameters that have a stable optimum across different network scales under the network parameterization. Naturally, this excludes hyperparameters that define the training scale, which other hyperparameters are transferred across. As shown in the paper, hyperparameters with a strong regularization effect, such as dropout probability and weight decay, were empirically found not to transfer well.

---

[4]We note that some optimizers recommend 10 times the number of hyperparameters as samples alone for the initial design before the actual optimization starts.

### 2.2.2   Gradient-Based AutoML

Instead of having an outer loop training an ML model with different configurations over and over again, it would be desirable to learn the hyperparameters on the fly while training the ML model. This would be specifically interesting for LLMs when we can afford only one training run. Although gradient-based optimization is possible both for HPO (Maclaurin et al., 2015; Luketina et al., 2016; Franceschi et al., 2017; MacKay et al., 2019; Lorraine et al., 2020) and neural architecture search w.r.t. individual cells of the network (Liu et al., 2019a; Elsken et al., 2019), these approaches struggle so far with robustness and scaling to large networks such as LLMs.

### 2.2.3   Meta-Learning When and How to Adjust Training

If we think about how human experts train LLMs, they use check-pointing s.t. they can intervene if the training threatens to fail, e.g., changing the learning rate accordingly. Overall, this manual strategy resembles dynamic configuration approaches (Adriaensen et al., 2022) that meta-learn how to adjust the hyperparameter configurations while the training of the model is ongoing. Going even one step further would lead to learning the learning process itself (Andrychowicz et al., 2016; Chen et al., 2022). Since this approach requires an offline meta-learning for obtaining the corresponding meta-policy (e.g., learned with RL), it is an open problem how to scale it up to LLMs where the collection of meta-learning and evaluations of the meta-policy is not trivially feasible.

## 2.3   Challenge II: A Multitude of Different Stages

As we illustrate in Figure 1, the LLM lifecycle comprises different stages and each of them has different objectives, subjects and hyperparameters. That makes a holistic AutoML approach very hard and, perhaps, even impossible. Although it is in principle possible to tune complex pipelines of predictive models (Wachsmuth et al., 2013; Feurer et al., 2015a; M. Wever & Hüllermeier, 2018; Olson & Moore, 2019), it would be too expensive to tune all stages of the LLM training jointly. Moreover, they even do not follow the same metrics for the objectives. Therefore, we usually tune them independently and discuss them step by step in the following.

### 2.3.1   Pre-Training

Hyperparameter Optimization for pre-training is not only expensive, as previously discussed, but also spans a variety of very different types of hyperparameters. Selecting data sources for pre-training is a subtle but crucial choice to this end. It affects the domain-specific and general knowledge encoded into the LLM, while also impacting its conversational and reasoning abilities (Zhao et al., 2023a). Additionally, data pre-processing decisions impact the data quality and, in turn, affect the downstream performance. As with all text-based models, tokenization (Schuster & Nakajima, 2012; Sennrich et al., 2016; Kudo, 2018) plays a key role in the training data representation. Subsequently, the intricacies of the backbone architecture of the LLM must be decided. Transformer-based models (Vaswani et al., 2017) are the avant-garde in language models. Designing such a transformer model comprises several crucial architectural choices such as a range of different encoder-decoder architectures (Vaswani et al., 2017; Lewis et al., 2020), decoder architectures (Brown et al., 2020; Zhang et al., 2022) and encoder architectures (Devlin et al., 2019; Liu et al., 2019b). Similarly, a key configuration option is the choice of self-supervised training strategies (Lewis et al., 2020). Besides, choices regarding normalization (Ba et al., 2016; Xiong et al., 2020), activation functions (Hendrycks & Gimpel, 2016; Shazeer, 2020), or positional encoding (Su et al., 2022), as well as training settings such as the optimizer (Kingma & Ba, 2015; Loshchilov & Hutter, 2018; Shazeer & Stern, 2018), learning rate schedule (Loshchilov & Hutter, 2017), and batch size have to be made.

### 2.3.2   Fine-Tuning

As fine-tuning approaches can be categorized into groups following different learning paradigms, corresponding challenges also differ according to the learning paradigm as we outline below.

**Supervised Task-Specific Fine-Tuning**   Task-specific fine-tuning can be regarded a standard supervised-learning stage based on input-output pairs (with or without instructions). That is, LLMs can still be fine-tuned to specific tasks without instruction tuning, and since the realization of the two is different, the application of AutoML differs in practice. AutoML for supervised fine-tuning in principle could follow the same approaches as extensively studied in the AutoML community since it provides a clear supervised training signal and is feasible computation-wise. Interesting questions arise with respect to neural architecture search to which we will come back later. One of the core challenges associated with supervised (task-specific) fine-tuning is that it can undo some of the alignment achieved via alignment fine-tuning (Qi et al., 2023) such that the two stages are ideally considered jointly when performing HPO.

**Instruction Tuning**   With instruction tuning being a specific type of generalized supervised task-specific fine-tuning (Wei et al., 2022) that prepares LLMs for prompting, similar challenges as for supervised task-specific fine-tuning arise when performing AutoML for this stage. In particular, this is the case since instruction tuning is also a form of supervised fine-tuning generalized to multiple tasks and thus, AutoML approaches for supervised learning can be used. While optimizing instruction tuning using AutoML boils down to the techniques already available for supervised learning problems, a few additional challenges arise.

First, curating the necessary task-specific data in sufficient quality and quantity is cumbersome and potentially labor-intensive. To alleviate the burdensome quantity issue, trained LLMs can be prompted using templates as a data extraction module (Zhang et al., 2023c). The quality of the extracted dataset, however, adheres to prompt engineering and LLM inference hyperparameters located in the pre-processing pipeline. They can be subjected to AutoML methods.

Second, oftentimes tasks share an inherent structure or knowledge as in coding tasks for code summarization or code completion. Exploiting the similarities using multi-task learning can facilitate improved (generalization) performance, robustness, data efficiency and data imbalance across tasks (Liu et al., 2023). With the ability to generate ample datasets and related tasks at arguably significant cost, task selection may become an issue. Kung et al. (2023) resort to active learning in order to solve it, but AutoML could also be invoked to assist in this selection.

A third challenge arises from the multi-task nature of the tuning process as the cost function optimized by an AutoML approach has to be able to capture the performance across multiple tasks. While this problem is prevalent in the related field of algorithm configuration (Schede et al., 2022), it is less so in AutoML. Nevertheless, corresponding approaches exist (Perrone et al., 2018; Law et al., 2019; Li et al., 2022). In general, AutoML can help with finding the right hyperparameter values for instruction tuning that are described by Wei et al. (2022) such as the number of gradient steps, the batch size, the optimizer or the learning rate.

**Alignment Fine-Tuning**   Alignment fine-tuning is usually performed via reinforcement learning, more precisely RLHF (Fernandes et al., 2023). Although RLHF can be seen as a form of instruction tuning, we dedicate a separate paragraph to alignment tuning and RLHF here, as they are complicated problems from an AutoML perspective due to their dependability on reinforcement learning (Eimer et al., 2023).[5] Unfortunately, RL is not well studied from an AutoML perspective. There are still many open questions, such as properties of hyperparameter landscapes in RL (Mohan et al., 2023), sound evaluation protocols (Eimer et al., 2023), stability of training due to the non-stationary learning task and non-deterministic data collection on the fly, especially in online and on-policy learning. The nonstationarity in the training data introduces noise to the performance observations for AutoML systems and increases the risk of overfitting the tuning process to a few random seeds (Eimer et al., 2023).

Automated Reinforcement Learning (AutoRL) (Parker-Holder et al., 2022) aims to address these issues through techniques such as hyperparameter optimization for RL (Li et al., 2019; Parker-Holder et al., 2020; Wan et al., 2022), learned optimizers (Lan et al., 2023), neural architecture search (Wan et al., 2022) and dynamic configurations (Adriaensen et al., 2022). Most of these considerations of AutoRL translate to the RL components of LLMs; thus, corresponding methods might be a suitable choice. However, at the current

---

[5]We note that there are recent approaches aimed at replacing reinforcement learning with supervised approaches, such as classification losses (Rafailov et al., 2023).

point in time, appropriate tuning of RL for LLM is even more understudied than AutoRL for traditional RL benchmarks such as control or games.

A crucial design decision is the RL algorithm itself. PPO (Schulman et al., 2017) and A2C (Mnih et al., 2016) are currently common choices for RLHF. However, in principle, the scalar nature of the reward in the RL optimization stage of LLM alignment allows seamless integration of many existing RL algorithms. Thus, selecting RL algorithms based on the task at hand could leverage further potential here (Laroche & Feraud, 2018).

Most RLHF-specific design decisions related to the reward model (RM) are not used in standard RL and thus are not studied in the existing AutoRL literature. A crucial design decision for the RM is determining its size relative to the pre-trained LLM. There is no established standard for this, with current solutions being driven heuristically. For example, OpenAI uses an RM with 6 billion parameters for an LLM with 175 billion parameters, while Deepmind uses the same size for both the RM and the LLM (Lambert et al., 2022). This design decision could depend on the downstream task for which the fine-tuning is being performed and the multiple objectives the preference score aims to capture. Optimizing for optimal size ratio for RM with respect to the LLM is a configuration problem that could be sped up via learning-curve-based multi-fidelity methods (Klein et al., 2017; Jawed et al., 2021; Ruhkopf et al., 2023). Recent results on the positive impact of incorporating multiple reward models to produce a more expressive reward signal (Wu et al., 2023c) open up new avenues for methods that can utilize ensembling methodologies for task-specific fine-tuning architectures. Moreover, methods to iteratively update the RM and policy together (Lambert et al., 2022; Bai et al., 2022) could open the doors to complex dynamics for which AutoRL hyperparameter landscapes (Mohan et al., 2023) can be utilized for designing better optimizers and multi-fidelity methods.

Another departure from standard AutoRL is the policy update in RLHF, which is usually performed only on a subset of the weights due to the size of the LLM (Hu et al., 2022a; Glaese et al., 2022). This subset of weights for the update depends on multiple factors, including the complexity of concepts in the data, preference scores, and the size of the LLM. Methods for data generation through exploration and curriculum learning to adaptively select the best data for the policy update can serve particularly useful in this scenario (Jiang et al., 2023). Techniques from multi-objective optimization could be further used to balance data quality, policy updates and number of parameters to update.

### 2.3.3 Inference

Inference queries imply forward passes through billion-parameter models, leading to high deployment costs that can be computationally as well as ecologically costly. The former is particularly important since a multitude of LLMs fine-tuned to a variety of tasks serve large communities of users. As a consequence, mitigating these costs and maximizing the utility for the users should be the prime objective of this stage and results in a difficult multi-objective optimization problem. Automated pruning techniques (Chen et al., 2020; Wang et al., 2020) can help to reduce this cost. In addition, mixed precision (Shen et al., 2020; Dettmers et al., 2022), which uses 16 bits (or even less) and 32 bits to represent floating point numbers, reduces memory usage and accelerates computations in training as well as inference (Yuan & Agaian, 2023). Similarly, adjusting the maximum number of generated tokens, i.e., the length of a response or the number of responses, in cases where the user asks for multiple ones, can help to reduce the cost but might harm the user benefit. Moreover, tuning hyperparameters that affect the randomness of the generated text, such as temperature or top-k adjustments, may increase the utility but naturally can impact the expected number of queries needed to achieve a desired output. Notably, advanced decoding strategies (Zhao et al., 2023a) including repeated prompting and templates or chain-of-thought related concepts (Besta et al., 2023; Ning et al., 2023) may provide noticeable improvements. However, automatically optimizing inference via means of AutoML is still an open challenge.

Nevertheless, first works in the area of automated prompt engineering exist. Shin et al. (2020) demonstrate that (approximate) gradient-based optimization can lead to prompts that lead to better results than hand-designed prompts. Similarly, Zhou et al. (2023) show that good prompts can be found in an automated fashion via an iterative interplay between several language models where one LLM suggests prompts or adaptations to prompts and another LLM rates these prompts. In general, prompt engineering is spurred by

the phenomenon of in-context learning(Brown et al., 2020) allowing LLMs to adjust their behavior through conditioning on input-output examples as part of a prompt without any changes to the underlying network weights. As such, work in the area of automated prompt engineering can also be seen as work in the area of in-context learning.

Wang et al. (2023a) take a first step towards applying AutoML for optimizing LLM inference by leveraging BlendSearch (Wang et al., 2021) and proposing a cost-based pruning strategy to optimize the inference hyperparameters of LLMs. This demonstrates that AutoML can indeed be used to optimize the inference stage of the LLM lifecycle.

On a more general level, knowledge distillation can be used to improve the inference speed of LLMs by creating smaller (student) LLMs with a similar performance to the original one (teacher) (Gu et al., 2023; Hsieh et al., 2023). AutoML approaches can be used here as well to optimize the hyperparameters of the corresponding knowledge distillation algorithms. For example, the MiniLLM approach suggested by Gu et al. (2023) leverages a gradient descent algorithm together with an advanced clipping strategy, both of which have hyperparameters, which can be optimized. Similarly, the step-by-step distillation by Hsieh et al. (2023) has several hyperparameters that can be optimized with AutoML approaches. Nevertheless, this can, in principle, be seen as yet another stage requiring a different AutoML setup compared to optimizing the inference pipeline itself as one can build an inference pipeline even around the student LLM obtained from the distillation process.

## 2.4 Challenge III: The Multitude of Performance Indicators

Eventually, we aim at obtaining a well-performing LLM system. A best practice for AutoML is to optimize the final performance of a system (or at least a very well-correlated proxy metric) to avoid any misalignment between the AutoML process and the important metrics after deployment of the system. However, it is not easy to answer what performance exactly entails and how it is quantified. This has several reasons: (i) Standard machine learning already comes with many possible performance metrics, and choosing the correct ones involves assessing their importance, which depends on the given application. For example, besides accuracy, the community considers inference time for high-throughput, memory, and energy consumption for edge devices. In the context of multimodal LLMs, various additional metrics may play a crucial role, such as image-text alignment (Xu et al., 2018; Grimal et al., 2023) or adversarial robustness (Zhao et al., 2023b). Multi-objective AutoML allows optimizing for several of these performance metrics (Morales-Hernández et al., 2021; Karl et al., 2023). (ii) While training the base model, the downstream task is not known, but the base model needs to be as general as possible. This implies that we do not know the final performance metric in earlier training stages but have to rely on the capabilities of the pre-trained model regarding its performance after fine-tuning (which will take place at some later point). (iii) In view of the prevalence of LLMs and the direct interaction with users, it is of utmost importance to consider the issue of bias and its implications (Kumar et al., 2023).

Considering the importance of the latter, let us discuss decreasing the bias of language model output via fine-tuning with AutoML in more detail. While language models themselves can be used as data generators for debiased data (Schick et al., 2021; Hernandez et al., 2023), as well as pre-defined sentence templates (Liang et al., 2020), AutoML can assist in determining the amount of additional data necessary as well as the kind of data that should be generated (data for removing bias, neutralizing representations, or equalizing them). Debiasing can also be interleaved with task objective fine-tuning (Saravanan et al., 2023) considering that the duration and amount of data used in both phases are important hyperparameters for AutoML methods to target. Gira et al. (2022) have shown that it is possible to fine-tune only a subset of model weights in order to achieve less biased outcomes – though selecting the correct parameters to tune is crucial for performance. Nevertheless, AutoML systems can only assist in training fairer models; human input is required for centering values like fairness in the whole training pipeline (Bender et al., 2021; Weerts et al., 2023).

Overall, as our elaboration shows, the topic of quantifying how good an LLM is for a specific use case is a complicated topic, which has also been discussed intensively in the literature. For example, Liang et al. (2023) demonstrate that different metrics are of different importance in different LLM use cases. Similarly,

Dehghani et al. (2022) elaborate on how detrimental the partial reporting of metrics can be for drawing final conclusions.

## 2.5 Challenge IV: Combination of Different Learning Paradigms and Modalities

Closely related to Challenge II and III, current AutoML packages are not well prepared for requirements for tuning the entire LLM training process because they commonly consider only a single learning paradigm at once. Training of LLMs is particularly challenging since it combines stages of self-supervised learning, supervised learning, and even reinforcement learning. This implies that we need separate design and configuration spaces for each stage while ultimately aiming at jointly optimizing all of them (Hutter, 2022). So far, configuration spaces for supervised classification learning already consider tenths or more than a hundred design decisions (Feurer et al., 2015b; Zimmer et al., 2021; Feurer et al., 2022). However, so far, it is unknown how we would jointly optimize the entire pipeline since this poses the sub-challenges of (i) jointly optimizing potentially hundreds of design decisions with (ii) several performance indicators on (iii) an unknown distribution of downstream tasks. Ways to go forward with this would entail studying the importance and interactions between design decisions to find feasible and potentially independent subsets of them, development of proxy performance signals or multi-objective optimization and optimization across many possible downstream tasks.

Furthermore, multi-modal LLMs are increasingly recognized for their potential in capturing diverse data types such as audio or images, and it is yet to be seen how this multi-modality will influence the optimization of the entire pipeline. Most likely, this will impact how we apply multi-fidelity approaches for efficient AutoML, e.g., how to choose informative fidelity types such as data subsets from different modalities and partial training of different components. Furthermore, this will further blow up the possible configuration space since all data modalities typically come with their own special design options.

## 2.6 Challenge V: Determining Neural Architectures for LLMs

Choosing an appropriate neural architecture is a crucial, but non-trivial step in designing state-of-the-art deep neural networks, including LLMs (White et al., 2023). Currently, this is mostly done manually by a human expert (Hutter, 2022). In contrast to handcrafted discrete architectural choices, Neural Architecture Search (NAS) aims at alleviating this manual effort while offering large flexibility in the choices powered by (partial) automation (Elsken et al., 2019). However, current NAS methods have yet to find new innovative state-of-the-art architectures or parts thereof in a way that self-attention was found by human experts (Vaswani et al., 2017). As such, they can be of great help in optimizing certain stages of the LLM lifecycle, but cannot be applied off-the-shelf without, along other things, a well-designed and potentially over-time adapting search space containing suitable architectural choices – let alone the computational challenges discussed earlier.

Recently, first efforts for tailoring NAS methods towards transformer-based models (Chitty-Venkata et al., 2022) to meet the specific needs of LLMs, have been made. For instance, (So et al., 2021) present a search strategy designed for transformer language models, utilizing a search space of TensorFlow programs and evolution to search for models. AutoBERT-Zero (Gao et al., 2021) specifically focuses on BERT models, introducing an inter-layer search space containing self-attention and convolution operations to provide flexibility in learning global and local dependencies at different layers, and proposing an operation-priority evolution strategy. Designed for vision transformers, Autoformer (Chen et al., 2021) utilizes a supernet training strategy called weight entanglement, entangling the weights of different blocks in the same layers during supernet training, and performs an evolutionary search over the supernets to find promising transformers.

There are also some special characteristics that we take into account for applying NAS for fine-tuning. As Aghajanyan et al. (2021) point out, fewer parameters are required for fine-tuning than for pre-training, and according to Lee et al. (2019), only some of the layers actually need fine-tuning. Therefore, we would benefit from intelligent methods to select the layers requiring fine-tuning and the subset of learnable model parameters that hold the relevant encoding to the subsequent task. For example, simply tuning a random sample of the model parameters (Aghajanyan et al., 2021) or adding adapter layers after the attention layers and only tuning those (Houlsby et al., 2019) effectively reduces the number of learnable model parameters

for fine-tuning. Another strategy in this regard is Low-Rank adaption, which freezes the weights from pre-training and introduces a correction term to them, which substantially reduces the number of parameters to train (Hu et al., 2022a). Leveraging AutoML methods for selecting and tuning the right subset of learnable parameters or layers and appropriately configuring the additional architecture introduced may prove valuable for downstream performance as well as parameter efficiency.

First empirical results show that NAS is indeed a promising direction for supporting the fine-tuning of language models (Chitty-Venkata et al., 2022). For instance, AdaBERT (Chen et al., 2020) uses NAS to automatically compress BERT for a specific task and Mahabadi et al. (2021) train a hypernetwork of adapter layers with shared parameters. Given its similarity to One-Shot NAS (Bender et al., 2018; Brock et al., 2018; Shi et al., 2020), which uses a super-network, methods to train the hyper- or super-network might be transferable to optimizing LLM fine-tuning.

When it comes to scaling models, parallelization of the base model training is an essential aspect and introduces new challenges in the design process. Parallelization induces architectural and hyperparameter design changes in the model, such as determining the optimal placement of batch normalization or residual connections to ensure stability and affect convergence speed and overall performance (Shoeybi et al., 2019). Acknowledging the significant impact on and necessity of parallelization for LLMs, it becomes crucial for NAS approaches to be parallel-aware.

Benchmarks for NAS sped up the development of new NAS algorithms and are predominant in the NAS literature since vast amounts of architectures were already evaluated (Ying et al., 2019; Dong & Yang, 2020; Mehta et al., 2022) and further extended by surrogate interpolations to even more architectures (Zela et al., 2022). However, we are only aware of a single benchmark of NAS for NLP (Klyuchnikov et al., 2022) that allows efficient benchmarking of different NAS approaches by evaluating about 14k different architectures. Unfortunately, these architectures are based on RNNs and not on modern transformers. Therefore, it is an open challenge how to provide a reproducible and efficient benchmark of NAS for LLMs to the community. For a general overview of desirable properties of NAS benchmarks, we refer to Lindauer & Hutter (2020).

# 3 LLMs for AutoML

At the moment it seems that LLMs have the potential to disrupt our society from a variety of angles, for example, in education (Kasneci et al., 2023), medicine (Alberts et al., 2023), programming (Dakhel et al., 2023), or in law (Noonan, 2023). As we illustrate in Figure 2 and elaborate in the following sections, we expect that LLMs will also disrupt AutoML from various angles:

(i) Interacting with complex systems such as an AutoML system is often challenging for non-expert users. The remarkable NLP capabilities of LLMs offer the opportunity to fundamentally redesign how humans interact with AutoML systems both from the point of view of setting them up and interpreting their output.

(ii) To unleash their full potential, AutoML systems have to be configured adequately often requiring an expert. The knowledge distillation capabilities of LLMs offer the opportunity to suggest a good initial configuration of an AutoML system for a specific problem at hand.

(iii) AutoML systems leverage several sub-components such as a neural performance predictor in many NAS tools. As first work shows, these components can be replaced by LLMs acting as meta-learned versions of the corresponding components.

## 3.1 Opportunity I: Improving Human-Machine-Interaction with LLMs

As natural language has always been the cornerstone of human communication, advances in NLP gave rise to an increasing amount of chatbots in various applications over the last decade (Adamopoulou & Moussiades, 2020). Until recently, however, most of these chatbots have been rather limited in their capabilities. As ChatGPT (OpenAI, 2022) shows, LLMs have the potential to alleviate this situation by allowing for significantly more

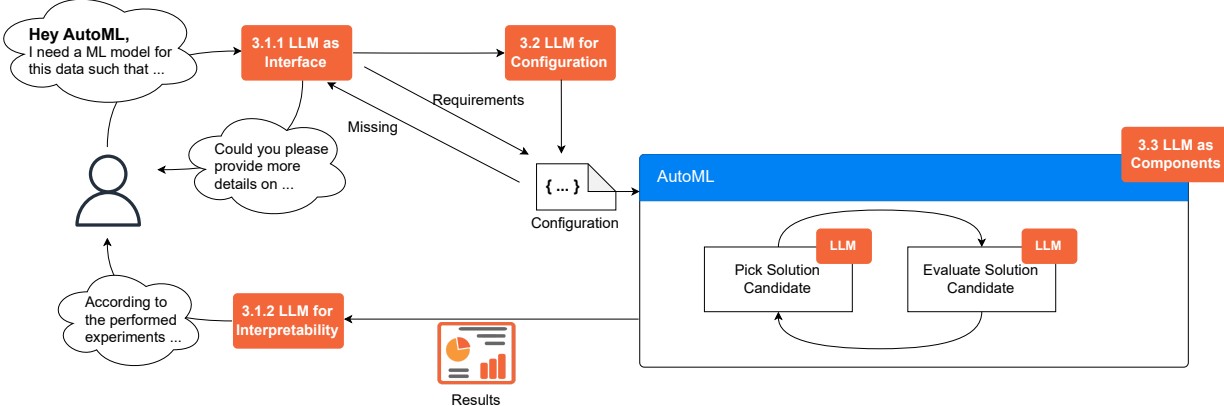

Figure 2: Overview of options where LLMs can be integrated into the AutoML process.

powerful chatbots and better textual interaction with a user. In the context of AutoML, we foresee two promising directions: Leveraging LLMs (i) as a user-friendly interface to AutoML, and (ii) to improve the interpretability of the AutoML process.

### 3.1.1 Opportunity I.a: LLMs as an Interface to AutoML

The original promise of AutoML was that it could automate some tasks of a data scientist to large degrees and thus democratize machine learning by allowing domain experts with little to no ML knowledge to apply ML to their domain. However, despite their success, many current AutoML tools were not built around the user, but rather around algorithmic ideas. In particular, most of these tools allow for very limited interaction with a user in practice. This is also reflected in the reluctance of many researchers to use AutoML tools (van der Blom et al., 2021). As a consequence, parts of the community have pushed towards a more human-centered AutoML process aimed at supporting the data scientist such that they can work more efficiently (Lindauer & Tornede, 2022; Pfisterer et al., 2019). Correspondingly, AutoML can be seen to have two main target groups: (i) Domain experts with little ML knowledge who want to apply off-the-shelf ML to their problem, and (ii) ML experts who want to improve their workflow with automated tools that keep them in the loop. Right now, most AutoML tools require coding or at least some technical understanding and, in terms of usability, target the second group much more than the first one.

LLMs enable us to fundamentally rethink how people interact with AutoML systems and, in particular, help us design powerful interactive text-based interfaces such as chatbots. These can iteratively extract the requirements of a user across a conversation and, in the background, configure an AutoML system correspondingly (see also Sec. 3.2) based on ML best practices and knowledge about optimization runs on similar datasets encoded in the LLM. In particular, such interactive systems can simplify many of the complicated design decisions affecting the AutoML process. For example, choosing an appropriate metric for optimization can be challenging for a non-expert, but might be possible with a competent interactive chatbot asking questions which guide the user to choosing the correct one. Naturally, this also bears the danger of wrong AutoML configurations (see Sec. 4).

Parts of this vision of conversational AutoML assistants are already a reality with AI-based coding assistant tools such as GitHub Copilot (Friedman, 2021). They can generate and suggest code to run AutoML with the contextual information that the users give. Thereby, AI-based coding assistants already assist users in finding concrete AutoML instantiations that best fit the requirement and the devices at hand. However, we envision assistants much more tailored to the needs of ML practitioners.

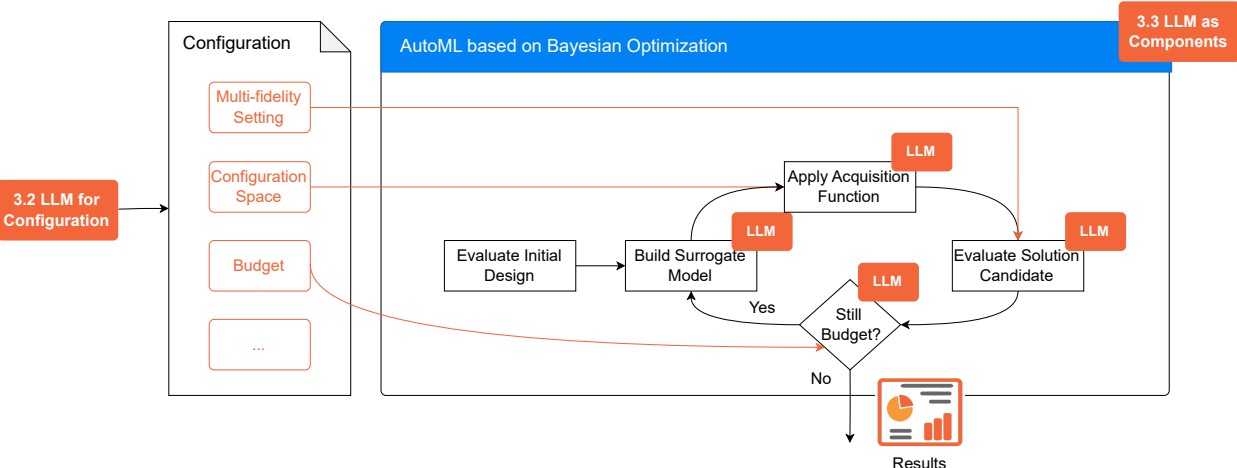

Figure 3: Visualization of the potential of LLMs for the configuration of AutoML (Section 3.2) and LLMs as components of AutoML systems (Section 3.3) at the example of a AutoML process based on Bayesian Optimization.

### 3.1.2 Opportunity I.b: Interpretability of the AutoML Process

There has been a recent rise in methods trying to contribute to the aforementioned human-centered AutoML paradigm (Pfisterer et al., 2019; Lindauer & Tornede, 2022; Moosbauer et al., 2021; 2022b; Segel et al., 2023) by proposing ideas to improve the interactivity with the user and the interpretability of the AutoML process. However, many of these works adapt rather classic interpretable machine learning methods to the AutoML setting, whose results might remain rather complicated to understand for non-experts and do not provide any textual, easy-to-understand explanations.

LLMs have the potential to significantly increase the user-friendliness of those interpretations by elaborating on them in the form of text. In particular, an LLM initialized with the history of evaluated configurations or pipelines, e.g. a run history from an optimizer such as SMAC (Lindauer et al., 2022), Hyperopt (Komer et al., 2014) or BoTorch (Balandat et al., 2020), as context could help generate a textual optimization report elaborating on the final AutoML result and details of the process itself. Ideally, the LLM could additionally be contextualized with results of several ixAutoML methods as a strong foundation for the report. This may even include images, such as partial dependence plots (Moosbauer et al., 2021), as multimodal LLMs allow to process images as well (Li et al., 2023; Zhang et al., 2023b). If the user has questions about the report or elements not covered by the report, the LLM could also be used as part of a chatbot to answer those questions. Although there is an ongoing discussion in the community on what constitutes a good explanation in general, textual explanations seem to be highly trusted by users (Gilpin et al., 2018).

### 3.2 Opportunity II: LLMs for Configuring AutoML

To fully shine, an AutoML system must be set up correctly, including a suitable system configuration for the task at hand. Going even further, selecting among various AutoML tools (Thornton et al., 2013; Feurer et al., 2015a; Akiba et al., 2019; Jin et al., 2019; Erickson et al., 2020; Zimmer et al., 2021) can be important depending on the problem and hardware at hand. Although there exists work in the direction of removing the burden of selecting and configuring an AutoML system (Feurer & Hutter, 2018; Tornede et al., 2022; Feurer et al., 2022; Moosbauer et al., 2022a), setting up an AutoML system without an expert can still be challenging. LLMs offer a great opportunity to further improve this situation by suggesting a good configuration for the task at hand. Below and in Figure 3, we outline several configuration options, which are often very important but difficult to choose correctly, even for experts.

The vast majority of AutoML tools require the configuration of a search space of candidates from which solutions can be drawn. Depending on the concrete application, this usually spans several numerical, categorical, and ordinal hyperparameters – often with dependencies between them. The concrete instantiation of this search space is crucial to find a well-performing pipeline quickly but is also hard to set up even for an expert. In particular, which kind of solutions (e.g. pipelines with or without preprocessing) are suitable, or which hyperparameters to tune and their concrete domains, are important to configure correctly. For example, choosing the domain of a hyperparameter to be small might benefit the search speed, but also bears the danger of missing the truly optimal value. Until now, this problem is either solved by an expert carefully configuring the AutoML tool, by approaches that adaptively adjust the search space during the optimization process (Wistuba et al., 2015; Nguyen et al., 2019; Hu et al., 2022b; 2020; Chen et al., 2019) or by integrating human expert knowledge as prior information guiding the search (Souza et al., 2021; Hvarfner et al., 2022; Mallik et al., 2022).

Similarly important – especially from a Green AutoML perspective (Tornede et al., 2023) – is the question of how long to run the AutoML process. Choosing a too-long runtime might waste both time and resources, while too-short runtimes bear the danger of missing good solutions. LLMs could provide first settings for maximum runtimes of AutoML tools based on the experience from other AutoML practitioners. Going even one step further, we could even leverage the knowledge encoded in an LLM to check whether the AutoML run has already converged or whether further tuning could still be promising. Recent work by Makarova et al. (2022) has shown that such an automatic termination is in principle possible, even without LLMs, and allows for considerable resource savings. In principle, we could even imagine that given a sufficient amount of AutoML studies with details on the configuration space being tuned, an LLM could also learn how much AutoML budget (e.g., in terms of evaluated configurations) is typically used for different configuration spaces to achieve a strong result.

Lastly, LLMs could be used to configure the use of multi-fidelity approaches, such as Hyperband (Li et al., 2018). As recent work shows, the performance of multi-fidelity approaches is influenced by the choice of the fidelity types (Deng et al., 2022) and the minimal/maximal amount of budget (Bohdal et al., 2023). Correspondingly, choosing these correctly is crucial but also hard in practice, making an automated suggestion very helpful.

### 3.3 Opportunity III: LLMs as Components of AutoML Systems

Most AutoML systems are complex tools with a plethora of sub-systems and components that suit different purposes, such as estimating the performance of a pipeline (White et al., 2021), estimating the runtime of a pipeline (Mohr et al., 2021), or choosing the next pipeline to evaluate. LLMs offer the opportunity to replace many of these sub-systems as a meta-learned version thereof. Below and in Figure 3 we outline and showcase several of such opportunities.

Almost all AutoML systems leverage some form of solution candidate selection strategy that selects the next candidate to be evaluated such as an acquisition function in BO-based systems. The knowledge modeling capabilities of LLMs and their access to tremendous amounts of meta-data about ML and AutoML runs offer the opportunity to replace these mostly hand-designed selection strategies with a meta-learned version in the form of an LLM. First works in these directions are promising: GPT-NAS (Yu et al., 2023a) leverages a GPT model to predict (parts) of a neural architecture, i.e., a solution candidate based on an encoding which is used for evolving architectures. GENIUS (Zheng et al., 2023) even goes a step further and replaces the whole architecture suggestion step with GPT-4. It iteratively prompts GPT-4 for an architecture, which it evaluates, and re-prompts GPT-4 with the performance asking for a better architecture. This process is continued until a stopping criterion is reached. Similarly, EvoPrompting (Chen et al., 2023) leverages an LLM to implement the crossover and mutation operator in an evolutionary NAS approach. Likewise, Nasir et al. (2023) suggest to leverage LLMs to generate individuals in an evolutionary NAS approach.

Moreover, the knowledge encoded in LLMs can also be used in feature engineering, as recently demonstrated. CAAFE (Hollmann et al., 2023b) is a feature engineering method for tabular datasets that leverages an LLM to iteratively propose additional code for feature engineering based on their descriptions and feedback back

the evaluation of this code piece to the LLMs. This contributes beyond AutoML to the ultimate goal of automated data science (Bie et al., 2022).

Furthermore, most AutoML tools iteratively evaluate new solution candidates by training and validating. Naturally, this is a time-consuming process, especially if the training time of the solution candidate is extremely long, such as in deep learning. For this reason, some systems leverage meta-learned performance estimators such as meta-learned surrogate models in Bayesian optimization (Vanschoren, 2019) or neural performance predictors in NAS (White et al., 2021) which can replace some of the evaluations. Once again, LLMs offer a great opportunity to serve as a special form of meta-learned replacement based on knowledge extracted from large amount of unstructured data, which is not accessible to standard meta-learned approaches. They can also serve as a basis to generate training data for simpler performance/training time estimators or surrogate models and potentially also replace the latter. Chen et al. (2022) have demonstrated that their OptFormer approach is able to learn both a competitive surrogate model and an acquisition function from the optimization history of Google's Vizier (Song et al., 2022) platform. Similarly, Liu et al. (2024) highlight that their approach LLMABO can be used to improve a variety of AutoML components such as warmstarting, surrogate modeling and candidate sampling.

Going even further, both Zhang et al. (2023d) and Zhang et al. (2023a) suggest AutoML-GPT and MLCopilot, respectively, which fully work as a zero-shot AutoML tool on their own. Given a textual problem description by the user and a knowledge base in the background, they suggest a pipeline and/or training procedure to achieve good performance on the specific problem. The problem itself can vary, but contains at least a description of the dataset a model is sought for, some form of search space description as a source of the model to be sought and a description on how performance is to be assessed. The search space can be a concrete class of models or even a concrete architecture such that the LLM needs to find a complete pipeline or only good hyperparameters (Zhang et al., 2023d). Note that these systems never evaluate a single ML pipeline, but, in the case of AutoML-GPT, only use LLMs to simulate the entire AutoML process. Tsai et al. (2023) take a slightly different approach by using an LLM to generate code to perform concrete AutoML tasks in an iterative process which is controlled by another LLM receiving the execution results and adjusting the prompt to the code LLM correspondingly. Similarly, Aliro (Choi et al., 2023) provide an AutoML tool tailored towards medical data where LLMs are used in the background to quickly generate code for user and case-specific analyses of the data.

The knowledge encoded in LLMs can, in principle, be derived from a multitude of data sources, including but not limited to academic papers, data science and ML competitions (e.g., Kaggle), ML benchmark platforms (e.g., OpenML (Casalicchio et al., 2019)), AutoML benchmarks, or AutoML runs with their corresponding performances. The increasing availability of open-source data and code may equip LLMs with the capability to grasp intricate relationships between architectural choices, hyperparameters, performance outcomes, or runtimes. Such data can be used to tailor LLMs to specific tasks, as done for example in GPT-NAS (Yu et al., 2023a), which is pre-trained on the NAS-Bench-101 dataset (Ying et al., 2019) and fine-tuned on a dataset adopting 36 presently prevalent neural architectures. However, the successes of LLMs in performing NAS (Zheng et al., 2023) or suggesting a whole AutoML pipeline (Zhang et al., 2023a;d), even without fine-tuning, suggests that the models can generalize their understanding from the wealth of information available.

All of the examples above crucially depend on how the prompts for the LLM are designed and how the knowledge gained so far is added as context to the prompts. For example, EvoPrompting (Chen et al., 2023) adds the code of all previously evaluated architectures together with their evaluation results as an annotation to the prompt as context. Naturally, this can quickly lead to very large prompts, which can be challenging for current LLMs, although recent work tries to alleviate the prompt size limitation (Yu et al., 2023b).

## 4   Risks

Besides the numerous valuable ways to integrate LLMs and AutoML, the combination poses some risks. Below, we elaborate five risks combining LLMs and AutoML:

(i) Using LLMs for configuring AutoML requires extracting meta-knowledge about AutoML tasks to be fed into the LLMs. Therefore, intensive and human-time-consuming prompt engineering is required to reliably extract that knowledge.

(ii) The usage of publicly available data to evaluate AutoML approaches configured via LLMs might be data snooping, as the LLM might have already seen the data during training. The detection of prior knowledge of an LLM about a given dataset becomes more and more important.

(iii) LLMs are known to hallucinate from time to time, presenting false facts due to the lack of fact-checking. This could be crucial when LLMs are used to configure AutoML as inappropriate configurations could be used.

(iv) The full-text interaction of LLMs seems to be quite trustworthy for laypeople, although the presented content might not be true at all. In combination with AutoML, the user might mistakenly get a positive impression of the results.

(v) Combining two resource-intensive research areas, i.e. LLMs and AutoML, will result in an even more resource-intensive research area. Thus, it is more important to be transparent about the resources used and find ways to be more efficient.

## 4.1 Risk I: Complicated Human-Interaction

We have leveraged the working assumption that LLMs trained on a large corpus of text also encode knowledge about AutoML tasks several times throughout this manuscript. Such knowledge could comprise a general understanding of what models perform well given a dataset description or which preprocessing is required for a specific task at hand. However, extracting such knowledge for the user in a reliable way depends on the specific prompt, leading to the task of prompt engineering for AutoML (Sorensen et al., 2022). Ideally, human-machine-interaction will make designing a concrete prompt obsolete by automatically constructing it in an automated fashion from a chat-like interaction with a user. However, this might only work to a certain extent and currently remains an open problem. Correspondingly, we risk that by leveraging LLMs for AutoML we only shift the barrier of entry from requiring coding and ML knowledge to prompt engineering knowledge.

## 4.2 Risk II: Evaluation

Any form of evaluation of an AutoML approach leveraging LLMs in some component needs to be performed with care. As LLMs are trained on extreme amounts of unstructured data, it is quite likely that they see many openly available ML datasets during training including corresponding test data or a condensed version thereof, such as a model trained on the data. Thus, evaluating an AutoML pipeline that uses an LLM on a dataset that was part of the training data of the underlying LLM is akin to data snooping (Kok, 1984). Correspondingly, the evaluation result would be strongly biased. We see two main strategies to circumvent this problem: First, we could evaluate it on datasets that are guaranteed not to have been seen by the LLM during training, such as data generated after the LLM has been trained (Faggioli et al., 2023). Naturally, this raises the question of how to get such private datasets, which are still freely accessible for other researchers, of realistic nature in order to compare approaches, but still were not used by the LLM in advance. Second, we could try to fine-tune an LLM to remove any knowledge about a dataset from its model (Yang et al., 2022). To achieve this, we would need to know how to detect prior knowledge of the LLMs about any used dataset or a condensed version thereof. Once such knowledge is detectable, we would need to find a way to reliably remove it via fine-tuning. However, at this point in time, there is no approach guaranteeing that prior knowledge about a dataset can be fully removed. Overall, the evaluation of an AutoML system featuring an LLM is far from trivial and requires new protocols.

## 4.3 Risk III: False Facts and Misuse

LLMs are known to generate output sounding confident but featuring hallucinated knowledge (Ji et al., 2023), which might be indistinguishable or at least hard to distinguish from facts in some cases. Correspondingly,

we may wonder whether any usage of LLMs within AutoML systems can lead to detrimental results, e.g., when misconfiguring the AutoML via an LLM by the user such that almost no runtime is granted, or the LLM takes decisions like choosing the search space which might be inappropriate for the given problem. A potential solution to this might be the integration with a knowledge graph (Hogan et al., 2022) tailored towards AutoML that can be used as a knowledge base to contextualize potential LLM suggestions. Moreover, we should consider filters or safeguards for degenerate solutions output by the LLM and resolve to predefined defaults in those cases.

In general, when integrating LLMs with AutoML systems, we should keep in mind the potential for wrong LLM output and how such output can be detected in a probabilistic approach. A big advantage over many other disciplines is that we can validate whether replies by the LLM are correct by simply running an experiment and comparing the experiment result output by the LLM and the result we obtain from the run (Zheng et al., 2023; Chen et al., 2023; Hollmann et al., 2023b).

### 4.4 Risk IV: Trust and Explanations

Although improving the human-machine-interaction of AutoML systems leveraging LLMs (Sec. 3.1) has the potential to democratize ML even more, it also bears dangers when more laypeople interact with such systems. For example, due to the full-text interaction with a user, there might be a lot of trust in the results, although the returned machine learning model might not be suitable for the given problem at all. Correspondingly, the attention of the user should be drawn to potential ambiguities in the interaction, and they should be made well aware of the assumptions the overall system makes due to the textual interaction. One way to address this issue could be to add further explanation approaches telling users how certain replies were generated (Deb et al., 2023).

### 4.5 Risk V: Resource Consumption

Both LLMs and AutoML are resource-intensive research areas on their own. Combining them could lead to even higher resource consumption, raising the question if spending these resources is worth it. In view of the growing number of fine-tuned LLMs for medical applications (Han et al., 2023; Wu et al., 2023a), it is quite likely that there will also be fine-tuned LLMs for specialized AutoML tasks. With the potential of easily generating more and more meta-knowledge for AutoML by simply evaluating more ML pipelines or hyperparameter configurations, it could even lead to a race of repeatedly fine-tuned tasks.[6] Nevertheless, the potentially rich meta-knowledge stored in these LLMs (see Section 3.2) could also lead to new efficiency gains in AutoML. Overall, in the spirit of Green AutoML (Tornede et al., 2023), it will be even more important to be transparent about the resources used when conducting research at this intersection, and to find ways to be more efficient on both sides.

## 5 Conclusion

LLMs already started to have big impact on AutoML, both how AutoML is designed for LLMs, and how LLMs can be leveraged for AutoML. Nevertheless, we are only at the beginning of a long journey to solve the many challenges and make use of the opportunities while considering the risks ahead of us. We strongly believe that we, as a community, will overcome those challenges and savour the remarkable opportunities that an integration of LLMs and AutoML offers.

The biggest challenge of all can be boiled down to one fact: The training and maintenance of the most capable LLMs requires immense resources, such that only a few groups worldwide are able to provide those systems. This implies several problems for the community: (i) We are not easily able to study how AutoML can be tailored toward training of LLM base models; (ii) We cannot easily check which data was used to train the models which bears risks in using and evaluating LLMs for AutoML; (iii) We cannot easily add safeguards against misuse of LLMs and AutoML. Correspondingly, it is even more problematic that the responsibility and control over LLMs lie in those few hands of mostly private companies. In view of this, we believe that

---

[6]We note that in contrast to real NLP tasks where human input is required, it is much easier to generate new examples of how ML pipelines or AutoML systems behave and perform.

one of the next steps has to be the development an open-source LLM with the specific goals of the AutoML in mind.

**Acknowledgments**

We would like to thank the wider AutoML community for various perspectives that have influenced this work. We would like to mention various discussions with the AutoML.org supergroup, the AutoML 2022 panel discussion on "AutoML in the age of large pretrained model" and breakout sessions at a small AutoML 2023 workshop in Paris on "AutoML & LLMs". Indeed, our paper's title was inspired by these events as well. In particular, we acknowledge (in alphabetic order): Steven Adriaensen, Deebadepta Dey, Carola Doerr, Katharina Eggensperger, Matthias Feurer, Neil Houlsby, Frank Hutter, Arjun Krishnakumar, Neeratyoy Mallik, Samuel Müller, Raghu Rajan, Elena Raponi, Zi Wang and Marc Zöller.

Alexander Tornede, Sarah Segel and Marius Lindauer acknowledge funding by the European Union (ERC, "ixAutoML", grant no.101041029). Views and opinions expressed are however those of the author(s) only and do not necessarily reflect those of the European Union or the European Research Council Executive Agency. Neither the European Union nor the granting authority can be held responsible for them. Tanja Tornede acknowledges financial support in the GreenAutoML4FAS project (no. 67KI32007A) and Daphne Theodorakopoulos in the ZuSiNa project (no. 67KI21009A), both funded by the German Federal Ministry of the Environment, Nature Conservation, Nuclear Safety and Consumer Protection. Tim Ruhkopf acknowledges financial support by the Federal Ministry for Economic Affairs and Energy of Germany in the project CoyPu (no. 01MK21007L).

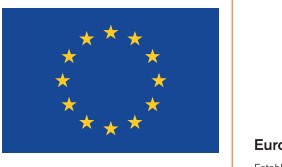

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
