# OpenReview forum: "AutoML in the Age of Large Language Models: Current Challenges, Future Opportunities and Risks"
_TMLR — Accepted by TMLR_

### Review · Reviewer_ieVj · 2023-08-05

**Summary Of Contributions:**

The paper presents a survey of existing literature at the boundaries of Auto ML and NLP (especially the development of large language model (LLM)). The paper is organized into suggesting how AutoML can aid the development of LLMs, and how LLMs can help Auto LM, and lastly discussing the risks and pitfalls involved in the process.

The intersection between AutoML and LLM development is a timely and important topic, thus I agree having a thoughtful survey can be useful for the scientific community. However, as is, I found this survey rather lacking in its contents. See strengths / weaknesses discussions below.

**Audience:**

Yes

**Broader Impact Concerns:**

I don't think this paper has broader impact concerns.

**Claims And Evidence:**

No

**Requested Changes:**

Overall, having a bit more concrete and systematic outline of existing work and suggestion for future work could be helpful.

===
* Section 2.3.4 -> there's rich work in decoding for LLMs such as nucelus sampling, contrastive decoding, etc. Probably worth mentioning it here?
* Section 2.5 -> transition to fair LM is a bit abrupt here... In some sense this is most relevant to the alignment phase of the LLM training?
* Section 2.1 -> When discussing the cost of pre-training base models, they simply mention *one* scenario of training GPT3. Might be worth motivating it with open sourced models and clearly listing out how much efficient different fidelity methods can be helpful.
Minor comments:
* I liked the section right before 2.5 where it discusses lack of benchmark for NAS for LLM -- but what are requirement for benchmark for NAS? Overall the survey will be more clear if there are more concrete suggestions for future work.
*  Section 2.5 -> Might be good to cite many efforts in NLP of capturing multiple performance indicators, such as Holistic Evaluation of Language Models (Liang et al, 2022) or The Efficiency Misnomer (Dehgani et al 2022).

**Strengths And Weaknesses:**

Strength:
* The paper is well organized.
* The topic is timely.

Weaknesses:
* The scope of the survey / targeted audience is unclear. Describing the scope of the survey clearly at the beginning might be helpful. Thesedays, LLM development is a expansive research area that spans hardware development, algorithm design, dataset curation, and so on. Some arguments are just about LLM in general, nothing particular about the auto ML + LLM (for example, section 4.2 and 4.3).
* Figure 1 / Section 2.1. / Section 2.3.2: I think what's labeled as "Supervised Fine-tuning" here is a bit confusing. When people do supervised fine-tuning for specific task, those are typically not followed by the alignment fine-tuning step. What's more common is instruction tuning (FINETUNED LANGUAGE MODELS ARE ZERO-SHOT LEARNERS (Wei et al ICLR 2022)), which contains a set of fine-tuning tasks. I think it might need some clarification...
* I found the writing to be somewhat verbose and a bit unclear/imprecise (e.g., when discussing Zhange et al in page 12, what do you mean by "good performance"? what do you mean by "knowledge about AutoML tasks" in Section 4.1 (page 13)).
* I have issues with section 3 and a bit concerned with many of the use cases suggested here.
  - 3.1.2 -> The inner workings of LLM remains mysterious, and just because they generate fluent text it does not mean they can be a reliable tool for explanation.  Indeed LLM will provide user-friendly explanation in the form of natural language, but how would it faithfully represent the auto ML process? I can see LLM working as a QA agent which reads and understand the documentation of auto ML process (as suggested by author), but I wouldn't call this "interpretability" which implies going beyond simply looking up of information.
 -  3.3. How would LLM can help AutoLM to find out best configuration? Would this work for HPO or NAS or both? How would LLM acquire such a specialized knowledge?

---

### Review · Reviewer_T8we · 2023-12-04

**Summary Of Contributions:**

This paper is half way between a survey and a position paper. It lays out opportunities, risks and challenges for work at the intersection of AutoML and Large Language Models (LLMs). In particular, it spells out various challenges AutoML faces before it can be effectively applied to LLMs, and in the other direction, it looks at ways that LLMs can be employed to improve the useability, effectiveness and sample efficiency of AutoML. It closes with a section outlining the risks that the combination of AutoML and LLMs may bring.

Please note that this paper is clearly a revision (much of it is in blue text, indicating changes with respect to a previous review), but I appear to have no access to that previous review. I have written my review based only on this latest version of the paper.

**Audience:**

Yes

**Broader Impact Concerns:**

The authors have done a good job outlining the various risks entailed by this line of research.

**Claims And Evidence:**

Yes

**Requested Changes:**

Any attempt to address my weakness above would likely improve the utility of the survey portion of Section 3 to none-AutoML researchers.

I found a few awkward sentences that could be addressed:

“the ordering among potentials configurations on low-fidelity approximations should be correlated with the maximum fidelity”: potentials -> potential

“its performance after the, at that point future, fine-tuning” -> this took me 5 passes to parse correctly. Simpler just to say, “its performance after fine-tuning”

**Strengths And Weaknesses:**

Strengths:

- This blend of survey and mission statement is really nice. The narrative gives each cited paper a clear role, and nicely answers the question of why one might want to dive deeper for each of the various topics it introduces.
- I especially liked the front half on AutoML for LLMs. Even if one has no interest in AutoML, thinking about AutoML forces you to think about how to do model design and hyper-parameter search in a principled, data-driven and efficient manner, and the authors have done a great job of outlining how that process has changed with LLMs. I found the discussion and pointers to related work relevant and inspiring.
- As someone who is not an AutoML practitioner, I was surprisingly engaged by the second half as well. Again, for anyone who does any amount of ML, stopping to reflect on how much of your job could be and may soon be automated is healthy and challenging.
- The paper is very self-aware. It understands the limitations of AutoML, and how these limitations can be addressed or compounded by engaging with LLMs. It is honest with itself about the lack of adoption of AutoML and actively looks for solutions.

Weaknesses:

- The portion on LLMs for AutoML (Sections 3.2 and 3.3) felt very optimistic to me. I would have liked more discussion on why we can expect LLMs to have knowledge about architectures, ML hyper-parameters and how well they are likely to perform in a particular ML scenario. Clearly, there have been some recent successes, given the cited works, but it would have been nice to have a clear discussion of why we think these successes are happening: is it from code data, ingesting papers and tutorial material on ML/autoML, or something else? (iPython notebooks?) As an example, I was left wondering why LLMs should excel at adjusting maximum runtime (last paragraph of page 12), or adjusting architectures based on performance descriptions (second paragraphs of 3.3). These decisions seem to relate code and data in a way that doesn’t naturally occur in most texts I can think of. I think the answers lie in the cited papers; if the authors could spell out the intuitions early in the discussion on LLMs for AutoML, I think it would be a valuable, small, change to their existing narrative.

---

> ### Author Response · Authors · 2023-12-22
> **Thank you for the positive assessment and constructive feedback**
>
> We thank the reviewer for their kind words and highly appreciate their helpful feedback and suggestions. In response to the feedback on the LLMs for AutoML Sections (3.2 and 3.3), we have incorporated a new paragraph in Section 3.3. (second-to-last)  to address the question of why LLMs are expected to possess knowledge about architectural choices, hyperparameters, performance outcomes, or runtimes. Furthermore, we extended the discussion in Section 3.2 on how and why LLMs could learn to terminate AutoML runs or predict a reasonable configuration budget.
>
> Furthermore, following the reviewer’s suggestions, we have revised the mentioned sentences to enhance clarity and readability.
> Please note that all changes of this revision are highlighted in red in the document such that they can be distinguished from the first revision (blue) and can be found easily for checking the rebuttal.

---

### Review · Reviewer_7wbM · 2023-12-04

**Summary Of Contributions:**

This paper discusses challenges, opportunities, and risks when AutoML and LLM are jointly applied. On the one hand, AutoML and LLM can help each other: AutoML can improve LLM automatically towards any specified objectives, such as efficiency and performance, and LLM can improve AutoML regarding interpretability, interactiability, and capability; on the other hand, however, the high computational cost, the complicated pretraining-instruction tuning-alignment pipeline, the large downstream application space and the hallucination issue of LLMs hinders its combination with AutoML. The authors elaborated on each point and discussed relevant studies and potential future research directions.

**Audience:**

Yes

**Broader Impact Concerns:**

This is mostly a survey paper. I didn't find any ethical issues.

**Claims And Evidence:**

No

**Requested Changes:**

The following concerns should be addressed.

1. Some contents are highly redundant which hurts readability, e.g. Sections 2.3 and 2.6 discuss almost the same topic.
2. While task-specific tuning is a standard practice in the era of BERT, Bart, and T5, it's not very popular for the current LLMs. To some extent, it could be regarded as a special form of instruction tuning where only one task is applied.
3. On page 4, the authors stated "We also note that the order of these fine-tuning stages is not fixed and can be changed." Could you please show me any references demonstrating that alignment tuning could be applied before instruction tuning
4. Also on page 4, the authors argue for RLHF that "A policy that takes the output of an LM as an input to produce language tokens as outputs." The policy model takes user query or prompt as input. From this statement, I'm not sure whether the authors understand RLHF and the alignment of LLMs.
5. In instruction tuning, one important topic is missing. Instruction tuning is crucial to elicit the model's instruction understanding capability, but which tasks should be included in the tuning and how to select and combine data for each task form a big challenge where AutoML may help.
6. When applying AutoML to RLHF, the authors should discuss challenges resulting from integrating AutoML with online and on-policy learning.
7. As LLMs are computationally inefficient at inference. Standard practice often adopts knowledge distillation that trains a smaller student model mimicking the behavior of the large teacher model. How AutoML helps the training and modeling of student models should be discussed.
8. In Section 2.3.3, the authors mentioned "Mixed precision training" for inference efficiency. If I understand correctly, it mainly improves training. How did it help inference?
9. Apart from text-based LLMs, research on multi-modal LLM including audio, video, and image modeling, becomes very popular, which should also be included in the discussion.
10. Since LLMs can be used to improve AutoML, the topic of how to train LLMs for AutoML should also be discussed apart from "how AutoML is designed for LLMs, and how LLMs can be leveraged for AutoML."

**Strengths And Weaknesses:**

Strengths:

This paper presents a timely survey for the study of AutoML and LLM, covering different aspects and pointing out several challenges and opportunities, which offers some insights and guidance for researchers in these areas.


Weaknesses:

While the combination of AutoML and LLM is intriguing, this paper suffers from several crucial issues. Particularly, the discussion of LLM should be significantly improved.

---

> ### Author Response · Authors · 2023-12-22
> **Thank you for your constructive feedback**
>
> We thank the reviewer for their detailed review. We highly appreciate the reviewer’s helpful feedback and suggestions. Please see our revision for how we addressed your comments. Furthermore, please note that all changes of this revision are highlighted in red in the document such that they can be distinguished from the first revision (blue) and can be found easily for checking the rebuttal.
>
> In particular, we made the following changes:
>
> 1. [Redundancy] According to the reviewer’s recommendation, we restructured the challenges in Section 2. We merged the content from Section 2.6 into 2.3 to have a concise overview and discussion of the different learning steps. We reordered the remaining challenges s.t. we avoid redundancies as much as possible. Nevertheless, we note that some statements have to be repeated such that we can discuss the various angles of possible challenges in applying AutoML in LLM in a consistent and self-contained way.
> 1. [Task-specific tuning and instruction tuning] We thank the reviewer for noting this. We agree with the reviewer’s general statement that fine-tuning seems to be less and less important, but optimizing a model toward one specific task may still be critical for specific applications or companies. Therefore, we believe that task-specific fine-tuning and instruction tuning are two different steps and thus should both be mentioned and discussed explicitly. This is also in particular interesting from the view of AutoML. We added a sentence at the beginning of Section 2.3.2 explaining why we kept the separation between task-specific fine-tuning and instruction tuning.
> 1. [Order of tuning steps] We agree that it is neither common nor straightforward to do alignment tuning before instruction tuning. We removed the sentence “We also note that the order of these fine-tuning stages is not fixed and can be changed.” in Section 2.1, as we acknowledge that it might introduce confusion, and clarified in several places in Section 2.
> 1. [RLHF] We agree that the wording regarding RLHF offers potential for improvement. We corrected the formulation in Section 2.1.
> 1. [Data for instruction tuning] We appreciate the reviewer's insightful comment on the challenges associated with task selection and data combination in the context of instruction tuning. In Section 2.3.2, we now address this aspect by highlighting challenges: Fine-Tuning and instruction tuning are augmented by AutoML for task-specific data extraction using other LLMs. We further motivate, as per the reviewer’s request, the necessity for task selection and the potential role of AutoML in it.
> 1. [AutoML for online and on-policy learning] We added a discussion in 2.3.2 under “Alignment Fine-Tuning”. For all details regarding AutoRL, we politely refer to the AutoRL survey referenced in our paper.
> 1. [Knowledge Distillation] We thank the reviewer for this feedback. As we understand current approaches to knowledge distillation for LLMs, this technique distills the knowledge relevant to a certain task or domain from a teacher network to train a student network. We did not find evidence for research focusing on knowledge distillation which transfers all the knowledge of the teacher LLM to a student LLM. We kindly ask the reviewer to provide a reference to us.
> 1. [Mixed Precision Training] We argue that mixed precision gives a speed-up in training and inference because it uses less memory and accelerates computations in general. In fact, it is common to use mixed-precision components also for binary neural networks that still include some components with 16bit or 32bit precision. Nevertheless, the overall inference performance of these is better than that of full precision networks. We updated this accordingly in Section 2.3.3.
> 1. [Multimodal LLMs] We appreciate the reviewer’s suggestion and have incorporated multimodal LLMs in the introduction, highlighting their emerging possibilities, as well as some notes on the challenges arising for AutoML, see the new Section 2.5. Additionally, we have integrated relevant information in Section 3.1.2 regarding their role in enhancing interpretability within the AutoML process.
> 1. [Training LLMs for AutoML] We thank the reviewer for raising the question of how LLMs can be trained specifically for AutoML tasks. We assume that this question is mainly related to the type of data that can be utilized for the training. We have incorporated a paragraph in Section 3.3 (second-to-last), emphasizing the availability of diverse data sources and how they may be used to equip LLMs with knowledge regarding AutoML tasks.

---

> > ### Comment · Reviewer_7wbM · 2024-01-05
> > **Thanks for the detailed response**
> >
> > Thanks for the response and paper updates, which address most of my concerns.
> >
> > Re broad knowledge distillation of LLMs, please check out the following recent papers:
> > 1. Gu et al., Knowledge Distillation of Large Language Models
> > 2. Hsieh et al., Distilling Step-by-Step! Outperforming Larger Language Models with Less Training Data and Smaller Model Sizes

---

> > > ### Author Response · Authors · 2024-01-11
> > >
> > > We kindly thank the reviewer for the additional references. Based on these, we added an additional paragraph (can be found in olive) at the end of Section 2.3 detailing the requested point 7 [Knowledge Distillation] of the reviewer. We hope that with this additional change, we have been able to dispel all of the reviewer's concerns.

---

### Decision · Action_Editor_XhK7 · 2024-02-07

**Recommendation:** Accept as is

**Comment:**

The revision produced by the authors addressed many of the original flaws and final manuscript is significantly improved. Reviewers found the paper accessible and well written, which is important for an survey paper such as this one.

**Audience:**

AutoML and LLMs are of general interest to TMLR.

**Claims And Evidence:**

The three reviewers vote for accepting the paper. While the paper is not perfect, all reviewers agreed that this survey constitutes a timely contribution and a timely discussion of the connections between AutoML and LLMs. Claims are supported and the authors provide useful insights how to improve LLMs through AutoML.